# Learning Self-Imitating Diverse Policies

**Tanmay Gangwani**
Dept. of Computer Science
UIUC
`gangwan2@uiuc.edu`

**Qiang Liu**
Dept. of Computer Science
UT Austin
`lqiang@cs.utexas.edu`

**Jian Peng**
Dept. of Computer Science
UIUC
`jianpeng@uiuc.edu`

## Abstract

The success of popular algorithms for deep reinforcement learning, such as policy-gradients and Q-learning, relies heavily on the availability of an informative reward signal at each timestep of the sequential decision-making process. When rewards are only sparsely available during an episode, or a rewarding feedback is provided only after episode termination, these algorithms perform sub-optimally due to the difficultly in credit assignment. Alternatively, trajectory-based policy optimization methods, such as cross-entropy method and evolution strategies, do not require per-timestep rewards, but have been found to suffer from high sample complexity by completing forgoing the temporal nature of the problem. Improving the efficiency of RL algorithms in real-world problems with sparse or episodic rewards is therefore a pressing need. In this work, we introduce a self-imitation learning algorithm that exploits and explores well in the sparse and episodic reward settings. We view each policy as a state-action visitation distribution and formulate policy optimization as a divergence minimization problem. We show that with Jensen-Shannon divergence, this divergence minimization problem can be reduced into a policy-gradient algorithm with shaped rewards learned from experience replays. Experimental results indicate that our algorithm works comparable to existing algorithms in environments with dense rewards, and significantly better in environments with sparse and episodic rewards. We then discuss limitations of self-imitation learning, and propose to solve them by using Stein variational policy gradient descent with the Jensen-Shannon kernel to learn multiple diverse policies. We demonstrate its effectiveness on a challenging variant of continuous-control MuJoCo locomotion tasks.

## 1 Introduction

Deep reinforcement learning (RL) has demonstrated significant applicability and superior performance in many problems outside the reach of traditional algorithms, such as computer and board games (Mnih et al., 2015; Silver et al., 2016), continuous control (Lillicrap et al., 2015), and robotics (Levine et al., 2016). Using deep neural networks as functional approximators, many classical RL algorithms have been shown to be very effective in solving sequential decision problems. For example, a policy that selects actions under certain state observation can be parameterized by a deep neural network that takes the current state observation as input and gives an action or a distribution over actions as output. Value functions that take both state observation and action as inputs and predict expected future reward can also be parameterized as neural networks. In order to optimize such neural networks, policy gradient methods (Mnih et al., 2016; Schulman et al., 2015; 2017a) and Q-learning algorithms (Mnih et al., 2015) capture the temporal structure of the sequential decision problem and decompose it to a supervised learning problem, guided by the immediate and discounted future reward from rollout data.

Unfortunately, when the reward signal becomes sparse or delayed, these RL algorithms may suffer from inferior performance and inefficient sample complexity, mainly due to the scarcity of the immediate supervision when training happens in single-timestep manner. This is known as the temporal credit assignment problem (Sutton, 1984). For instance, consider the Atari Montezuma's revenge game – a reward is received after collecting certain items or arriving at the final destination in the lowest level, while no reward is received as the agent is trying to reach these goals. The sparsity of the reward makes the neural network training very inefficient and also poses challenges in exploration. It is not hard to see that many of the real-world problems tend to be of the form where

rewards are either only sparsely available during an episode, or the rewards are episodic, meaning that a non-zero reward is only provided at the end of the trajectory or episode.

In addition to policy-gradient and Q-learning, alternative algorithms, such as those for global- or stochastic-optimization, have recently been studied for policy search. These algorithms do not decompose trajectories into individual timesteps, but instead apply zeroth-order finite-difference gradient or gradient-free methods to learn policies based on the cumulative rewards of the entire trajectory. Usually, trajectory samples are first generated by running the current policy and then the distribution of policy parameters is updated according to the trajectory-returns. The cross-entropy method (CEM, Rubinstein & Kroese (2016)) and evolution strategies (Salimans et al., 2017) are two nominal examples. Although their sample efficiency is often not comparable to the policy gradient methods when dense rewards are available from the environment, they are more widely applicable in the sparse or episodic reward settings as they are agnostic to task horizon, and only the trajectory-based cumulative reward is needed.

Our contribution is the introduction of a new algorithm based on policy-gradients, with the objective of achieving better performance than existing RL algorithms in sparse and episodic reward settings. Using the equivalence between the policy function and its state-action visitation distribution, we formulate policy optimization as a divergence minimization problem between the current policy's visitation and the distribution induced by a set of experience replay trajectories with high returns. We show that with the Jensen-Shannon divergence ($D_{JS}$), this divergence minimization problem can be reduced into a policy-gradient algorithm with shaped, dense rewards learned from these experience replays. This algorithm can be seen as *self-imitation learning*, in which the expert trajectories in the experience replays are self-generated by the agent during the course of learning, rather than using some external demonstrations. We combine the divergence minimization objective with the standard RL objective, and empirically show that the shaped, dense rewards significantly help in sparse and episodic settings by improving credit assignment. Following that, we qualitatively analyze the shortcomings of the self-imitation algorithm. Our second contribution is the application of Stein variational policy gradient (SVPG) with the Jensen-Shannon kernel to simultaneously learn multiple diverse policies. We demonstrate the benefits of this addition to the self-imitation framework by considering difficult exploration tasks with sparse and deceptive rewards.

**Related Works.** Divergence minimization has been used in various policy learning algorithms. Relative Entropy Policy Search (REPS) (Peters et al., 2010) restricts the loss of information between policy updates by constraining the KL-divergence between the state-action distribution of old and new policy. Policy search can also be formulated as an EM problem, leading to several interesting algorithms, such as RWR (Peters & Schaal, 2007) and PoWER (Kober & Peters, 2009). Here the M-step minimizes a KL-divergence between trajectory distributions, leading to an update rule which resembles return-weighted imitation learning. Please refer to Deisenroth et al. (2013) for a comprehensive exposition. MATL (Wulfmeier et al., 2017) uses adversarial training to bring state occupancy from a real and simulated agent close to each other for efficient transfer learning. In Guided Policy Search (GPS, Levine & Koltun (2013)), a parameterized policy is trained by constraining the divergence between the current policy and a controller learnt via trajectory optimization.

*Learning from Demonstrations (LfD).* The objective in LfD, or imitation learning, is to train a control policy to produce a trajectory distribution similar to the demonstrator. Approaches for self-driving cars (Bojarski et al., 2016) and drone manipulation (Ross et al., 2013) have used human-expert data, along with Behavioral Cloning algorithm to learn good control policies. Deep Q-learning has been combined with human demonstrations to achieve performance gains in Atari (Hester et al., 2017) and robotics tasks (Večerík et al., 2017; Nair et al., 2017). Human data has also been used in the maximum entropy IRL framework to learn cost functions under which the demonstrations are optimal (Finn et al., 2016). Ho & Ermon (2016) use the same framework to derive an imitation-learning algorithm (GAIL) which is motivated by minimizing the divergence between agent's rollouts and external expert demonstrations. Besides humans, other sources of expert supervision include planning-based approaches such as iLQR (Levine et al., 2016) and MCTS (Silver et al., 2016). Our algorithm departs from prior work in forgoing external supervision, and instead using the past experiences of the learner itself as demonstration data.

*Exploration and Diversity in RL.* Count-based exploration methods utilize state-action visitation counts $N(s, a)$, and award a bonus to rarely visited states (Strehl & Littman, 2008). In large state-spaces, approximation techniques (Tang et al., 2017), and estimation of pseudo-counts by learning

density models (Bellemare et al., 2016; Fu et al., 2017) has been researched. Intrinsic motivation has been shown to aid exploration, for instance by using information gain (Houthooft et al., 2016) or prediction error (Stadie et al., 2015) as a bonus. Hindsight Experience Replay (Andrychowicz et al., 2017) adds additional goals (and corresponding rewards) to a Q-learning algorithm. We also obtain additional rewards, but from a discriminator trained on past agent experiences, to accelerate a policy-gradient algorithm. Prior work has looked at training a diverse ensemble of agents with good exploratory skills (Liu et al., 2017; Conti et al., 2017; Florensa et al., 2017). To enjoy the benefits of diversity, we incorporate a modification of SVPG (Liu et al., 2017) in our final algorithm.

In very recent work, Oh et al. (2018) propose exploiting past good trajectories to drive exploration. Their algorithm buffers $(s, a)$ and the corresponding return for each transition in rolled trajectories, and reuses them for training if the stored return value is higher than the current state-value estimate. Our approach presents a different objective for self-imitation based on divergence-minimization. With this view, we learn shaped, dense rewards which are then used for policy optimization. We further improve the algorithm with SVPG. Reusing high-reward trajectories has also been explored for program synthesis and semantic parsing tasks (Liang et al., 2016; 2018; Abolafia et al., 2018).

## 2 MAIN METHODS

We start with a brief introduction to RL in Section 2.1, and then introduce our main algorithm of self-imitating learning in Section 2.2. Section 2.3 further extends our main method to learn multiple diverse policies using Stein variational policy gradient with Jensen-Shannon kernel.

### 2.1 REINFORCEMENT LEARNING BACKGROUND

A typical RL setting involves an environment modeled as a Markov Decision Process with an unknown system dynamics model $p(s_{t+1}|s_t, a_t)$ and an initial state distribution $p_0(s_0)$. An agent interacts sequentially with the environment in discrete time-steps using a policy $\pi$ which maps the an observation $s_t \in \mathcal{S}$ to either a single action $a_t$ (deterministic policy), or a distribution over the action space $\mathcal{A}$ (stochastic policy). We consider the scenario of stochastic policies over high-dimensional, continuous state and action spaces. The agent receives a per-step reward $r_t(s_t, a_t) \in \mathcal{R}$, and the RL objective involves maximization of the expected discounted sum of rewards, $\eta(\pi_\theta) = \mathbb{E}_{p_0, p, \pi}\left[\sum_{t=0}^{\infty} \gamma^t r(s_t, a_t)\right]$, where $\gamma \in (0, 1]$ is the discount factor. The action-value function is $Q^\pi(s_t, a_t) = \mathbb{E}_{p_0, p, \pi}\left[\sum_{t'=t}^{\infty} \gamma^{t'-t} r(s_{t'}, a_{t'})\right]$. We define the unnormalized $\gamma$-discounted state-visitation distribution for a policy $\pi$ by $\rho_\pi(s) = \sum_{t=0}^{\infty} \gamma^t P(s_t = s|\pi)$, where $P(s_t = s|\pi)$ is the probability of being in state $s$ at time $t$, when following policy $\pi$ and starting state $s_0 \sim p_0$. The expected policy return $\eta(\pi_\theta)$ can then be written as $\mathbb{E}_{\rho_\pi(s,a)}[r(s, a)]$, where $\rho_\pi(s, a) = \rho_\pi(s)\pi(a|s)$ is the state-action visitation distribution. Using the policy gradient theorem (Sutton et al., 2000), we can get the direction of ascent $\nabla_\theta \eta(\pi_\theta) = \mathbb{E}_{\rho_\pi(s,a)}\left[\nabla_\theta \log \pi_\theta(a|s) Q^\pi(s, a)\right]$.

### 2.2 POLICY OPTIMIZATION AS DIVERGENCE MINIMIZATION WITH SELF-IMITATION

Although the policy $\pi(a|s)$ is given as a conditional distribution, its behavior is better characterized by the corresponding state-action visitation distribution $\rho_\pi(s, a)$, which wraps the MDP dynamics and fully decides the expected return via $\eta(\pi) = \mathbb{E}_{\rho_\pi}[r(s, a)]$. Therefore, distance metrics on a policy $\pi$ should be defined with respect to the visitation distribution $\rho_\pi$, and the policy search should be viewed as finding policies with good visitation distributions $\rho_\pi$ that yield high reward. Suppose we have access to a good policy $\pi^*$, then it is natural to consider finding a $\pi$ such that its visitation distribution $\rho_\pi$ matches $\rho_{\pi^*}$. To do so, we can define a divergence measure $D(\rho_\pi, \rho_{\pi^*})$ that captures the similarity between two distributions, and minimize this divergence for policy improvement.

Assume there exists an expert policy $\pi_E$, such that policy optimization can be framed as minimizing the divergence $\min_\pi D(\rho_\pi, \rho_{\pi_E})$, that is, finding a policy $\pi$ to imitate $\pi_E$. In practice, however, we do not have access to any real guiding expert policy. Instead, we can maintain a selected subset $\mathcal{M}_E$ of highly-rewarded trajectories from the previous rollouts of policy $\pi$, and optimize the policy $\pi$ to minimize the divergence between $\rho_\pi$ and the empirical state-action pair distribution $\{(s_i, a_i)\}_{\mathcal{M}_E}$:

$$\min_\pi D(\rho_\pi, \{(s_i, a_i)\}_{\mathcal{M}_E}). \tag{1}$$

Since it is not always possible to explicitly formulate $\rho_\pi$ even with the exact functional form of $\pi$, we generate rollouts from $\pi$ in the environment and obtain an empirical distribution of $\rho_\pi$. To measure the divergence between two empirical distributions, we use the Jensen-Shannon divergence, with the following variational form (up to a constant shift) as exploited in GANs (Goodfellow et al., 2014):

$$D_{JS}(\rho_\pi, \rho_{\pi_E}) = \max_{d(s,a), d_E(s,a)} \widetilde{\mathbb{E}}_{\rho_\pi}[\log \frac{d(s,a)}{d(s,a) + d_E(s,a)}] + \widetilde{\mathbb{E}}_{\rho_{\pi_E}}[\log \frac{d_E(s,a)}{d(s,a) + d_E(s,a)}], \quad (2)$$

where $d(s,a)$ and $d_E(s,a)$ are empirical density estimators of $\rho_\pi$ and $\rho_{\pi_E}$, respectively. Under certain assumptions, we can obtain an approximate gradient of $D_{JS}$ w.r.t the policy parameters, thus enabling us to optimize the policy.

**Gradient Approximation:** *Let $\rho_\pi(s,a)$ and $\rho_{\pi_E}(s,a)$ be the state-action visitation distributions induced by two policies $\pi$ and $\pi_E$ respectively. Let $d_\pi$ and $d_{\pi_E}$ be the surrogates to $\rho_\pi$ and $\rho_{\pi_E}$, respectively, obtained by solving Equation 2. Then, if the policy $\pi$ is parameterized by $\theta$, the gradient of $D_{JS}(\rho_\pi, \rho_{\pi_E})$ with respect to policy parameters ($\theta$) can be approximated as*:

$$\nabla_\theta D_{JS}(\rho_\pi, \rho_{\pi_E}) \approx \widetilde{\mathbb{E}}_{\rho_\pi(s,a)}\big[\nabla_\theta \log \pi_\theta(a|s)\widetilde{Q}^\pi(s,a)\big],$$

$$\text{where } \widetilde{Q}^\pi(s_t, a_t) = \widetilde{\mathbb{E}}_{\rho_\pi(s,a)}\Big[\sum_{t'=t}^\infty \gamma^{t'-t}\log \frac{d_\pi(s_{t'}, a_{t'})}{d_\pi(s_{t'}, a_{t'}) + d_{\pi_E}(s_{t'}, a_{t'})}\Big].. \quad (3)$$

The derivation of the approximation and the underlying assumptions are in Appendix 5.1. Next, we introduce a simple and inexpensive approach to construct the replay memory $\mathcal{M}_E$ using high-return past experiences during training. In this way, $\rho_{\pi_E}$ can be seen as a mixture of deterministic policies, each representing a delta point mass distribution in the trajectory space or a finite discrete visitation distribution of state-action pairs. At each iteration, we apply the current policy $\pi_\theta$ to sample $b$ trajectories $\{\tau\}_1^b$. We hope to include in $\mathcal{M}_E$, the top-$k$ trajectories (or trajectories with returns above a threshold) generated thus far during the training process. For this, we use a priority-queue list for $\mathcal{M}_E$ which keeps the trajectories sorted according to the total trajectory reward. The reward for each newly sampled trajectory in $\{\tau\}_1^b$ is compared with the current threshold of the priority-queue, updating $\mathcal{M}_E$ accordingly. The frequency of updates is impacted by the exploration capabilities of the agent and the stochasticity in the environment. We find that simply sampling noisy actions from Gaussian policies is sufficient for several locomotion tasks (Section 3). To handle more challenging environments, in the next sub-section, we augment our policy optimization procedure to explicitly enhance exploration and produce an ensemble of diverse policies.

In the usual imitation learning framework, expert demonstrations of trajectories—from external sources—are available as the empirical distribution of $\rho_{\pi_E}$ of an expert policy $\pi_E$. In our approach, since the agent learns by treating its own good past experiences as the expert, we can view the algorithm as *self-imitation learning* from experience replay. As noted in Equation 3, the gradient estimator of $D_{JS}$ has a form similar to policy gradients, but for replacing the true reward function with per-timestep reward defined as $\log(d_\pi(s,a)/(d_\pi(s,a) + d_{\pi_E}(s,a)))$. Therefore, it is possible to interpolate the gradient of $D_{JS}$ and the standard policy gradient. We would highlight the benefit of this interpolation soon. The net gradient on the policy parameters is:

$$\nabla_\theta \eta(\pi_\theta) = (1-\nu)\mathbb{E}_{\rho_\pi(s,a)}\big[\nabla_\theta \log \pi_\theta(a|s)Q^r(s,a)\big] - \nu\nabla_\theta D_{JS}(\rho_\pi, \rho_{\pi_E}), \quad (4)$$

where $Q^r$ is the $Q$ function with true rewards, and $\pi_E$ is the mixture policy represented by the samples in $\mathcal{M}_E$. Let $r^\phi(s,a) = d_\pi(s,a)/[d_\pi(s,a) + d_{\pi_E}(s,a)]$. $r^\phi(s,a)$ can be computed using parameterized networks for densities $d_\pi$ and $d_{\pi_E}$, which are trained by solving the $D_{JS}$ optimization (Eq 2) using the current policy rollouts and $\mathcal{M}_E$, where $\phi$ includes the parameters for $d_\pi$ and $d_{\pi_E}$. Using Equation 3, the interpolated gradient can be further simplified to:

$$\nabla_\theta \eta(\pi_\theta) = \mathbb{E}_{\rho_\pi(s,a)}\Big[\nabla_\theta \log \pi_\theta(a|s)\big[(1-\nu)Q^r(s,a) + \nu Q^{r^\phi}(s,a)\big]\Big], \quad (5)$$

where $Q^{r^\phi}(s_t, a_t) = -\mathbb{E}_{p_0, p, \pi}\big[\sum_{t'=t}^\infty \gamma^{t'-t}\log r^\phi(s_{t'}, a_{t'})\big]$ is the $Q$ function calculated using $-\log r^\phi(s,a)$ as the reward. This reward is high in the regions of the $\mathcal{S} \times \mathcal{A}$ space frequented more by the expert than the learner, and low in regions visited more by the learner than the expert. The effective $Q$ in Equation 5 is therefore an interpolation between $Q^r$ obtained with true environment rewards, and $Q^{r^\phi}$ obtained with rewards which are implicitly shaped to guide the learner

towards expert behavior. In environments with sparse or deceptive rewards, where the signal from $Q^r$ is weak or sub-optimal, a higher weight on $Q^{r^\phi}$ enables successful learning by imitation. We show this empirically in our experiments. We further find that even in cases with dense environment rewards, the two gradient components can be successfully combined for policy optimization. The complete algorithm for self-imitation is outlined in Appendix 5.2 (Algorithm 1).

**Limitations of self-imitation.** We now elucidate some shortcomings of the self-imitation approach. Since the replay memory $\mathcal{M}_E$ is only constructed from the past training rollouts, the quality of the trajectories in $\mathcal{M}_E$ is hinged on good exploration by the agent. Consider a maze environment where the robot is only rewarded when it arrives at a goal $\mathcal{G}$ placed in a far-off corner. Unless the robot reaches $\mathcal{G}$ once, the trajectories in $\mathcal{M}_E$ always have a total reward of zero, and the learning signal from $Q^{r^\phi}$ is not useful. Secondly, self-imitation can lead to sub-optimal policies when there are local minima in the policy optimization landscape; for example, assume the maze has a second goal $\mathcal{G}'$ in the opposite direction of $\mathcal{G}$, but with a much smaller reward. With simple exploration, the agent may fill $\mathcal{M}_E$ with below-par trajectories leading to $\mathcal{G}'$, and the reinforcement from $Q^{r^\phi}$ would drive it further to $\mathcal{G}'$. Thirdly, stochasticity in the environment may make it difficult to recover the optimal policy just by imitating the past top-$k$ rollouts. For instance, in a 2-armed bandit problem with reward distributions Bernoulli (p) and Bernoulli (p+$\epsilon$), rollouts from both the arms get conflated in $\mathcal{M}_E$ during training with high probability, making it hard to imitate the action of picking the arm with the higher expected reward.

We propose to overcome these pitfalls by training an *ensemble* of self-imitating agents, which are explicitly encouraged to visit different, non-overlapping regions of the state-space. This helps to discover useful rewards in sparse settings, avoids deceptive reward traps, and in environments with reward-stochasticity like the 2-armed bandit, increases the probability of the optimal policy being present in the final trained ensemble. We detail the enhancements next.

## 2.3 Improving Exploration with Stein Variational Gradient

One approach to achieve better exploration in challenging cases like above is to simultaneously learn multiple diverse policies and enforce them to explore different parts of the high dimensional space. This can be achieved based on the recent work by Liu et al. (2017) on Stein variational policy gradient (SVPG). The idea of SVPG is to find an optimal distribution $q(\theta)$ over the policy parameters $\theta$ which maximizes the expected policy returns, along with an entropy regularization that enforces diversity on the parameter space, i.e.

$$\max_q \mathbb{E}_{\theta \sim q}[\eta(\theta)] + \alpha H(q).$$

Without a parametric assumption on $q$, this problem admits a challenging functional optimization problem. Stein variational gradient descent (SVGD, Liu & Wang (2016)) provides an efficient solution for solving this problem, by approximating $q$ with a delta measure $q = \sum_{i=1}^n \delta_{\theta_i}/n$, where $\{\theta_i\}_{i=1}^n$ is an ensemble of policies, and iteratively update $\{\theta_i\}$ with

$$\theta_i \leftarrow \theta_i + \epsilon \Delta \theta_i, \qquad \Delta \theta_i = \frac{1}{n} \sum_{j=1}^n \left[ \nabla_{\theta_j} \eta(\pi_{\theta_j}) k(\theta_j, \theta_i) + \alpha \nabla_{\theta_j} k(\theta_j, \theta_i) \right] \qquad (6)$$

where $k(\theta_j, \theta_i)$ is a positive definite kernel function. The first term in $\Delta \theta_i$ moves the policy to regions with high expected return (exploitation), while the second term creates a repulsion pressure between policies in the ensemble and encourages diversity (exploration). The choice of kernel is critical. Liu et al. (2017) used a simple Gaussian RBF kernel $k(\theta_j, \theta_i) = \exp(-\|\theta_j - \theta_i\|_2^2/h)$, with the bandwidth $h$ dynamically adapted. This, however, assumes a flat Euclidean distance between $\theta_j$ and $\theta_i$, ignoring the structure of the entities defined by them, which are probability distributions. A statistical distance, such as $D_{JS}$, serves as a better metric for comparing policies (Amari, 1998; Kakade, 2002). Motivated by this, we propose to improve SVPG using JS kernel $k(\theta_j, \theta_i) = \exp(-D_{JS}(\rho_{\pi_{\theta_j}}, \rho_{\pi_{\theta_i}})/T)$, where $\rho_{\pi_\theta}(s, a)$ is the state-action visitation distribution obtained by running policy $\pi_\theta$, and $T$ is the temperature. The second exploration term in SVPG involves the gradient of the kernel w.r.t policy parameters. With the JS kernel, this requires estimating gradient of $D_{JS}$, which as shown in Equation 3, can be obtained using policy gradients with an appropriately trained reward function.

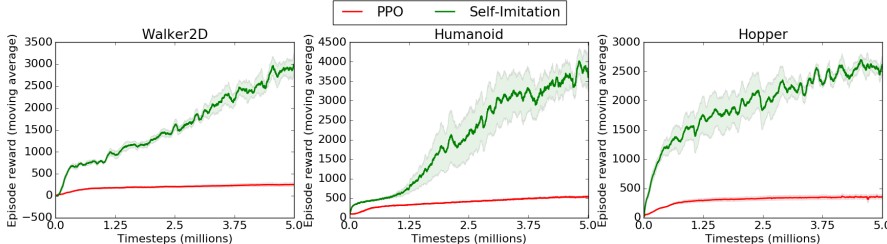

Figure 1: Learning curves for PPO and Self-Imitation on tasks with episodic rewards. Mean and standard-deviation over 5 random seeds is plotted.

Our full algorithm is summarized in Appendix 5.3 (Algorithm 2). In each iteration, we apply the SVPG gradient to each of the policies, where the $\nabla_\theta \eta(\pi_\theta)$ in Equation 6 is the interpolated gradient from self-imitation (Equation 5). We also utilize state-value function networks as baselines to reduce the variance in sampled policy-gradients.

## 3 EXPERIMENTS

Our goal in this section is to answer the following questions: 1) How does self-imitation fare against standard policy gradients under various reward distributions from the environment, namely episodic, noisy and dense? 2) How far does the SVPG exploration go in overcoming the limitations of self-imitation, such as susceptibility to local-minimas?

We benchmark high-dimensional, continuous-control locomotion tasks based on the MuJoCo physics simulator by extending the OpenAI Baselines (Dhariwal et al., 2017) framework. Our control policies ($\theta_i$) are modeled as unimodal Gaussians. All feed-forward networks have two layers of 64 hidden units each with tanh non-linearity. For policy-gradient, we use the clipped-surrogate based PPO algorithm (Schulman et al., 2017b). Further implementation details are in the Appendix.

| | Episodic rewards | | | | Noisy rewards Each $r_t$ suppressed w/ 90% prob. ($p_m = 0.9$) | | Noisy rewards Each $r_t$ suppressed w/ 50% prob. ($p_m = 0.5$) | | Dense rewards (Gym default) | |
|---|---|---|---|---|---|---|---|---|---|---|
| | $\nu = 0.8$ (SI) | $\nu = 0$ (PPO) | CEM | ES | $\nu = 0.8$ (SI) | $\nu = 0$ (PPO) | $\nu = 0.8$ (SI) | $\nu = 0$ (PPO) | $\nu = 0.8$ (SI) | $\nu = 0$ (PPO) |
| Walker | 2996 | 252 | 205 | $\approx 1200$ | 2276 | 2047 | 3049 | 3364 | 3263 | 3401 |
| Humanoid | 3602 | 532 | 426 | - | 4136 | 1159 | 4296 | 3145 | 3339 | 4149 |
| H-Standup ($\times 10^4$) | 18.1 | 4.4 | 9.6 | - | 14.3 | 11.4 | 16.3 | 9.8 | 17.2 | 10 |
| Hopper | 2618 | 354 | 97 | $\approx 1900$ | 2381 | 2264 | 2137 | 2132 | 2700 | 2252 |
| Swimmer | 173 | 21 | 17 | - | 52 | 37 | 127 | 56 | 106 | 68 |
| Invd.Pendulum | 8668 | 344 | 86 | $\approx 9000$ | 8744 | 8826 | 8926 | 8968 | 8989 | 8694 |

Table 1: Performance of PPO and Self-Imitation (SI) on tasks with episodic rewards, noisy rewards with masking probability $p_m$, and dense rewards. All runs use 5M timesteps of interaction with the environment. ES performance at 5M timesteps is taken from (Salimans et al., 2017). Missing entry denotes that we were unable to obtain the 5M timestep performance from the paper.

### 3.1 SELF-IMITATION WITH DIFFERENT REWARD DISTRIBUTIONS

We evaluate the performance of self-imitation with a single agent in this sub-section; combination with SVPG exploration for multiple agents is discussed in the next. We consider the locomotion tasks in OpenAI Gym under 3 separate reward distributions: **Dense** refers to the default reward function in Gym, which provides a reward for each simulation timestep. In **episodic** reward setting, rather than providing $r(s_t, a_t)$ at each timestep of an episode, we provide $\sum_t r(s_t, a_t)$ at the *last* timestep of the episode, and zero reward at other timesteps. This is the case for many practical settings where the reward function is hard to design, but scoring each trajectory, possibly by a human (Christiano et al., 2017), is feasible. In **noisy** reward setting, we probabilistically mask out each out each per-timestep reward $r(a_t, s_t)$ in an episode. Reward masking is done independently for every new episode, and therefore, the agent receives non-zero feedback at different—albeit only

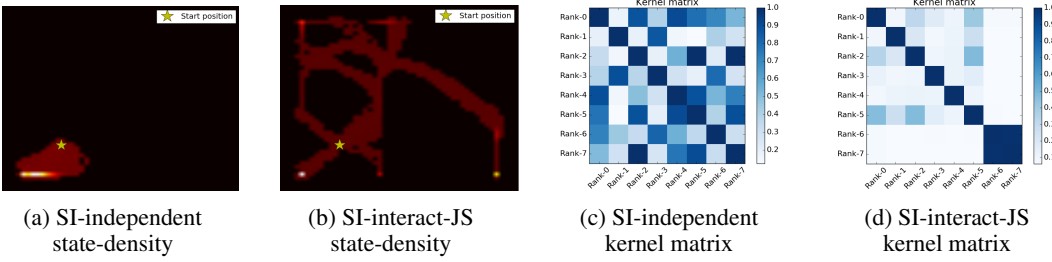

| (a) SI-independent state-density | (b) SI-interact-JS state-density | (c) SI-independent kernel matrix | (d) SI-interact-JS kernel matrix |

Figure 2: SI-independent and SI-interact-JS agents on Maze environment.

few—timesteps in different episodes. The probability of masking-out or suppressing the rewards is denoted by $p_m$.

In Figure 1, we plot the learning curves on three tasks with episodic rewards. Recall that $\nu$ is the hyper-parameter controlling the weight distribution between gradients with environment rewards and the gradients with shaped reward from $r^\phi$ (Equation 5). The baseline PPO agents use $\nu = 0$, meaning that the entire learning signal comes from the environment. We compare them with self-imitating (SI) agents using a constant value $\nu = 0.8$. The capacity of $\mathcal{M}_E$ is fixed at 10 trajectories. We didn't observe our method to be particularly sensitive to the choice of $\nu$ and the capacity value. For instance, $\nu = 1$ works equally well. Further ablation on these two hyper-parameters can be found in the Appendix.

In Figure 1, we see that the PPO agents are unable to make any tangible progress on these tasks with episodic rewards, possibly due to difficulty in credit assignment – the lumped rewards at the end of the episode can't be properly attributed to the individual state-action pairs during the episode. In case of Self-Imitation, the algorithm has access to the shaped rewards for each timestep, derived from the high-return trajectories in $\mathcal{M}_E$. This makes credit-assignment easier, leading to successful learning even for very high-dimensional control tasks such as Humanoid.

Table 1 summarizes the final performance, averaged over 5 runs with random seeds, under the various reward settings. For the noisy rewards, we compare performance with two different reward masking values - suppressing each reward $r(s_t, a_t)$ with 90% probability ($p_m = 0.9$), and with 50% probability ($p_m = 0.5$). The density of rewards increases across the reward settings from left to right in Table 1. We find that SI agents ($\nu = 0.8$) achieve higher average score than the baseline PPO agents ($\nu = 0$) in majority of the tasks for all the settings. This indicates that not only does self-imitation vastly help when the environment rewards are scant, it can readily be incorporated with the standard policy gradients via interpolation, for successful learning across reward settings. For completion, we include performance of CEM and ES since these algorithms depend only on the total trajectory rewards and don't exploit the temporal structure. CEM perform poorly in most of the cases. ES, while being able to solve the tasks, is sample-inefficient. We include ES performance from Salimans et al. (2017) after 5M timesteps of training for a fair comparison with our algorithm.

## 3.2 CHARACTERIZING ENSEMBLE OF DIVERSE SELF-IMITATING POLICIES

We now conduct experiments to show how self-imitation can lead to sub-optimal policies in certain cases, and how the SVPG objective, which trains an ensemble with an explicit $D_{JS}$ repulsion between policies, can improve performance.

**2D-Navigation.** Consider a simple Maze environment where the start location of the agent (blue particle) is shown in the figure on the right, along with two regions – the red region is closer to agent's starting location but has a per-timestep reward of only 1 point if the agent hovers over it; the green region is on the other side of the wall but has a per-timestep reward of 10 points. We run 8 independent, non-interacting, self-imitating (with $\nu = 0.8$) agents on this task. This ensemble is denoted as **SI-independent**. Figures 2a plots the state-visitation density for SI-independent after training, from which it is evident that the agents get trapped in the local minima. The red-region is relatively easily explored and trajectories leading to it fill the $\mathcal{M}_E$, causing sub-optimal imitation. We contrast this with an instantiation of our full

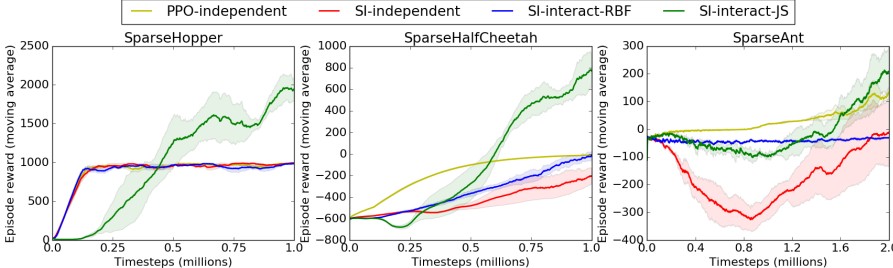

Figure 3: Learning curves for various ensembles on sparse locomotion tasks. Mean and standard-deviation over 3 random seeds are plotted.

algorithm, which is referred to as **SI-interact-JS**. It is composed of 8 self-imitating agents which share information for gradient calculation with the SVPG objective (Equation 6). The temperature $T = 0.5$ is held constant, and the weight on exploration-facilitating repulsion term ($\alpha$) is linearly decayed over time. Figure 2b depicts the state-visitation density for this ensemble. SI-interact-JS explores wider portions of the maze, with multiple agents reaching the green zone of high reward.

Figures 2c and 2d show the kernel matrices for the two ensembles after training. Cell $(i, j)$ in the matrix corresponds to the kernel value $k(\theta_i, \theta_j) = \exp(-JS(\rho_i, \rho_j)/T)$. For SI-independent, many darker cells indicate that policies are closer (low JS). For SI-interact-JS, which explicitly tries to decrease $k(\theta_i, \theta_j)$, the cells are noticeably lighter, indicating dissimilar policies (high JS). Behavior of PPO-independent ($\nu = 0$) is similar to SI-independent ($\nu = 0.8$) for the Maze task.

**Locomotion.** To explore the limitations of self-imitation in harder exploration problems in high-dimensional, continuous state-action spaces, we modify 3 MuJoCo tasks as follows – *Sparse-HalfCheetah*, *SparseHopper* and *SparseAnt* yield a forward velocity reward only when the center-of-mass of the corresponding bot is beyond a certain threshold distance. At all timesteps, there is an energy penalty to move the joints, and a survival bonus for bots that can fall over causing premature episode termination (Hopper, Ant). Figure 3 plots the performance of PPO-independent, SI-independent, SI-interact-JS and SI-interact-RBF (which uses RBF-kernel from Liu et al. (2017) instead of the JS-kernel) on the tasks. Each of these 4 algorithms is an ensemble of 8 agents using the same amount of simulation timesteps. The results are averaged over 3 separate runs, where for each run, the best agent from the ensemble after training is selected.

The SI-independent agents rely solely on action-space noise from the Gaussian policy parameterization to find high-return trajectories which are added to $\mathcal{M}_E$ as demonstrations. This is mostly inadequate or slow for sparse environments. Indeed, we find that all demonstrations in $\mathcal{M}_E$ for *SparseHopper* are with the bot standing upright (or tilted) and gathering only the survival bonus, as action-space noise alone can't discover hopping behavior. Similarly, for *SparseHalfCheetah*, $\mathcal{M}_E$ has trajectories with the bot haphazardly moving back and forth. On the other hand, in SI-interact-JS, the $D_{JS}$ repulsion term encourages the agents to be diverse and explore the state-space much more effectively. This leads to faster discovery of quality trajectories, which then provide good reinforcement through self-imitation, leading to higher overall score. SI-interact-RBF doesn't perform as well, suggesting that the JS-kernel is more formidable for exploration. PPO-independent gets stuck in the local optimum for *SparseHopper* and *SparseHalfCheetah* – the bots stand still after training, avoiding energy penalty. For *SparseAnt*, the bot can cross our preset distance threshold using only action-space noise, but learning is slow due to naïve exploration.

## 4 CONCLUSION AND FUTURE WORK

We approached policy optimization for deep RL from the perspective of JS-divergence minimization between state-action distributions of a policy and its own past good rollouts. This leads to a self-imitation algorithm which improves upon standard policy-gradient methods via the addition of a simple gradient term obtained from implicitly shaped dense rewards. We observe substantial performance gains over the baseline for high-dimensional, continuous-control tasks with episodic and noisy rewards. Further, we discuss the potential limitations of the self-imitation approach, and

propose ensemble training with the SVPG objective and JS-kernel as a solution. Through experimentation, we demonstrate the benefits of a self-imitating, diverse ensemble for efficient exploration and avoidance of local minima.

An interesting future work is improving our algorithm using the rich literature on exploration in RL. Since ours is a population-based exploration method, techniques for efficient *single agent* exploration can be readily combined with it. For instance, parameter-space noise or curiosity-driven exploration can be applied to each agent in the SI-interact-JS ensemble. Secondly, our algorithm for training diverse agents could be used more generally. In Appendix 5.6, we show preliminary results for two cases: a) hierarchical RL, where a diverse group of Swimmer bots is trained for downstream use in a complex Swimming+Gathering task; b) RL without environment rewards, relying solely on diversity as the optimization objective. Further investigation is left for future work.

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

## 5 APPENDIX

### 5.1 DERIVATION OF GRADIENT APPROXIMATION

Let $d_\pi^*(s, a)$ and $d_E^*(s, a)$ be the exact state-action densities for the current policy ($\pi_\theta$) and the expert, respectively. Therefore, by definition, we have (up to a constant shift):

$$D_{JS}(\rho_{\pi_\theta}, \rho_{\pi_E}) = \widetilde{\mathbb{E}}_{\rho_{\pi_\theta}}[\log \frac{d_\pi^*(s, a)}{d_\pi^*(s, a) + d_E^*(s, a)}] + \widetilde{\mathbb{E}}_{\rho_{\pi_E}}[\log \frac{d_E^*(s, a)}{d_\pi^*(s, a) + d_E^*(s, a)}]$$

Now, $d_\pi^*(s, a)$ is a local surrogate to $\rho_{\pi_\theta}(s, a)$. By approximating it to be constant in an $\epsilon-$ball neighborhood around $\theta$, we get the following after taking gradient of the above equation w.r.t $\theta$:

$$\nabla_\theta D_{JS}(\rho_{\pi_\theta}, \rho_{\pi_E}) \approx \nabla_\theta \widetilde{\mathbb{E}}_{\rho_{\pi_\theta}} [\log \underbrace{\frac{d_\pi^*(s, a)}{d_\pi^*(s, a) + d_E^*(s, a)}}_{r(s, a)}] + 0$$

$$= \widetilde{\mathbb{E}}_{\rho_{\pi_\theta}(s, a)} \big[\nabla_\theta \log \pi_\theta(a|s) \widetilde{Q}^\pi(s, a)\big],$$

$$\text{where } \widetilde{Q}^\pi(s_t, a_t) = \widetilde{\mathbb{E}}_{\rho_{\pi_\theta}(s, a)} \big[\sum_{t'=t}^\infty \gamma^{t'-t} r(s_{t'}, a_{t'})\big] \quad \square$$

The last step follows directly from the policy gradient theorem (Sutton et al., 2000). Since we do not have the exact densities $d_\pi^*(s, a)$ and $d_E^*(s, a)$, we substitute them with the optimized density estimators $d_\pi(s, a)$ and $d_E(s, a)$ from the maximization in Equation 2 for computing $D_{JS}$. This gives us the gradient approximation mentioned in Section 2.2. A similar approximation is also used by Ho & Ermon (2016) for Generative Adversarial Imitation Learning (GAIL).

## 5.2 ALGORITHM FOR SELF-IMITATION

Notation:
$\theta$ = Policy parameters
$\phi$ = Discriminator parameters
$r(s, a)$ = Environment reward

---

**Algorithm 1:**

---

1  $\theta, \phi \sim$ initial parameters
2  $\mathcal{M}_E \leftarrow$ empty replay memory
3  **for** *each iteration* **do**
4      Generate batch of trajectories $\{\tau\}_1^b$ with two rewards for each transition: $r_1 = r(s, a)$ and
       $r_2 = -\log r^\phi(s, a)$
5      Update $\mathcal{M}_E$ using priory queue threshold
    `/* Update policy θ */`
6      **for** *each minibatch* **do**
7          Calculate $g_1 = \nabla_\theta \eta^{r_1}(\pi_\theta)$ with PPO objective using $r_1$ reward
8          Calculate $g_2 = \nabla_\theta \eta^{r_2}(\pi_\theta)$ with PPO objective using $r_2$ reward
9          Update $\theta$ with $(1 - \nu)g_1 + \nu g_2$ using ADAM
10     **end**
    `/* Update self-imitation discriminator φ */`
11     **for** *each epoch* **do**
12         $s_1 \leftarrow$ Sample mini-batch of (s,a) from $\mathcal{M}_E$
13         $s_2 \leftarrow$ Sample mini-batch of (s,a) from $\{\tau\}_1^b$
14         Update $\phi$ with log-loss objective using $s_1, s_2$
15     **end**
16 **end**

---

### 5.3 ALGORITHM FOR SELF-IMITATING DIVERSE POLICIES

Notation:
$\theta_i$ = Policy parameters for rank $i$
$\phi_i$ = Self-imitation discriminator parameters for rank $i$
$\psi_i$ = Empirical density network parameters for rank $i$

---

**Algorithm 2:**

---

```
/* This is run for every rank i ∈ 1...n */
```

1   $\theta_i, \phi_i, \psi_i \sim$ some initial distributions
2   $\mathcal{M}_E \leftarrow$ empty replay memory local to rank $i$
3   $k(i,j) \leftarrow 0, \forall j \neq i$

4   **for** *each iteration* **do**
5      Generate batch of trajectories $\{\tau_i\}_1^b$
6      Update $\mathcal{M}_E$ using priory queue threshold

     `/* Update policy θ_i */`
7      **for** *each minibatch* **do**
8         Calculate $\nabla_{\theta_i} \eta(\pi_{\theta_i})$ using self-imitation (as in Algorithm 1)
9         MPI send: $\nabla_{\theta_i} \eta(\pi_{\theta_i})$ to other ranks
10        MPI recv: $\nabla_{\theta_j} \eta(\pi_{\theta_j})$ from other ranks
11        Calculate $\nabla_{\theta_i} k(i,j)$ using $\psi_i, \psi_j$
12        Use $k(i,j)$ and lines **8, 10, 11** in SVPG to get $\Delta\theta_i$
13        Update $\theta_i$ with $\Delta\theta_i$ using ADAM
14      **end**

     `/* Update self-imitation discriminator φ_i */`
15      **for** *each epoch* **do**
16        $s_1 \leftarrow$ Sample mini-batch of (s,a) from $\mathcal{M}_E$
17        $s_2 \leftarrow$ Sample mini-batch of (s,a) from $\{\tau_i\}_1^b$
18        Update $\phi_i$ with log-loss objective using $s_1, s_2$
19      **end**

     `/* Update state-action visitation network ψ_i */`
20      MPI send: $\psi_i$ to other ranks
21      MPI send: $\{\tau_i\}_1^b$ to other ranks
22      MPI recv: $\psi_j$ from other ranks
23      MPI recv: $\{\tau_j\}_1^b$ from other ranks
24      Update $\psi_i$ with log-loss objective using $\psi_j, \{\tau_i\}_1^b, \{\tau_j\}_1^b$
25      Update $k(i,j)$
26  **end**

---

## 5.4 Ablation Studies

We show the sensitivity of self-imitation to $\nu$ and the capacity of $\mathcal{M}_E$, denoted by $C$. The experiments in this subsection are done on Humanoid and Hopper tasks with episodic rewards. The tables show the average performance over 5 random seeds. For ablation on $\nu$, $C$ is fixed at 10; for ablation on $C$, $\nu$ is fixed at 0.8. With episodic rewards, a higher value of $\nu$ helps boost performance since the RL signal from the environment is weak. With $\nu = 0.8$, there isn't a single best choice for $C$, though all values of $C$ give better results than baseline PPO ($\nu = 0$).

|  | Humanoid | Hopper |
|---|---|---|
| $\nu = 0$ | 532 | 354 |
| $\nu = 0.2$ | 395 | 481 |
| $\nu = 0.5$ | 810 | 645 |
| $\nu = 0.8$ | 3602 | 2618 |
| $\nu = 1$ | 3891 | 2633 |

|  | Humanoid | Hopper |
|---|---|---|
| $C = 1$ | 2861 | 1736 |
| $C = 5$ | 2946 | 2415 |
| $C = 10$ | 3602 | 2618 |
| $C = 25$ | 2667 | 1624 |
| $C = 50$ | 4159 | 2301 |

## 5.5 Hyperparameters

- Horizon (T) = 1000 (locomotion), 250 (Maze), 5000 (Swimming+Gathering)
- Discount ($\gamma$) = 0.99
- GAE parameter ($\lambda$) = 0.95
- PPO internal epochs = 5
- PPO learning rate = 1e-4
- PPO mini-batch = 64

## 5.6 Leveraging Diverse Policies

The diversity-promoting $D_{JS}$ repulsion can be used for various other purposes apart from aiding exploration in the sparse environments considered thus far. First, we consider the paradigm of hierarchical reinforcement learning wherein multiple sub-policies (or skills) are managed by a high-level policy, which chooses the most apt sub-policy to execute at any given time. In Figure 4, we use the Swimmer environment from Gym and show that diverse skills (movements) can be acquired in a pre-training phase when $D_{JS}$ repulsion is used. The skills can then be used in a difficult downstream task. During pre-training with SVPG, exploitation is done with policy-gradients calculated using the norm of the velocity as dense rewards, while the exploration term uses the JS-kernel. As before, we compare an ensemble of 8 interacting agents with 8 independent agents. Figures 4a and 4b depict the paths taken by the Swimmer after training with independent and interacting agents, respectively. The latter exhibit variety. Figure 4c is the downstream task of Swimming+Gathering (Duan et al., 2016) where the bot has to swim and collect the green dots, whilst avoiding the red ones. The utility of pre-training a diverse ensemble is shown in Figure 4d, which plots the performance on this task while training a higher-level categorical manager policy ($|\mathcal{A}| = 8$).

Diversity can sometimes also help in learning a skill without any rewards from the environment, as observed by Eysenbach et al. (2018) in recent work. We consider a Hopper task with no rewards, but we do require weak supervision in form of the length of each trajectory $L$. Using policy-gradient

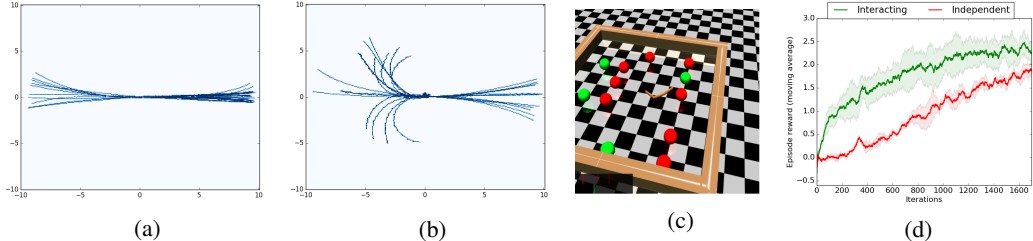

(a)       (b)       (c)       (d)

Figure 4: Using diverse agents for hierarchical reinforcement learning. (a) Independent agents paths. (b) Interacting agents paths. (c) Swimming+Gathering task. (d) Performance of manager policy with two different pre-trained ensembles as sub-policies.

with $L$ as reward and $D_{JS}$ repulsion, we see the emergence of hopping behavior within an ensemble of 8 interacting agents. Videos of the skills acquired can be found here [1].

## 5.7 PERFORMANCE ON MORE MUJOCO TASKS

| | Episodic rewards | | Noisy rewards Each $r_t$ suppressed w/ 90% prob. ($p_m = 0.9$) | | Noisy rewards Each $r_t$ suppressed w/ 50% prob. ($p_m = 0.5$) | | Dense rewards (Gym default) | |
|---|---|---|---|---|---|---|---|---|
| | $\nu = 0.8$ (SI) | $\nu = 0$ (PPO) | $\nu = 0.8$ (SI) | $\nu = 0$ (PPO) | $\nu = 0.8$ (SI) | $\nu = 0$ (PPO) | $\nu = 0.8$ (SI) | $\nu = 0$ (PPO) |
| Half-Cheetah | 3686 | -1572 | 3378 | 1670 | 4574 | 2374 | 4878 | 2422 |
| Reacher | -12 | -12 | -12 | -10 | -6 | -6 | -5 | -5 |
| Inv. Pendulum | 977 | 53 | 993 | 999 | 978 | 988 | 969 | 992 |

Table 2: Extension of Table 1 from Section 3. All runs use 5M timesteps of interaction with the environment.

## 5.8 ADDITIONAL DETAILS ON SVPG EXPLORATION WITH JS-KERNEL

### 5.8.1 SVPG FORMULATION

Let the policy parameters be parameterized by $\theta$. To achieve diverse, high-return policies, we seek to obtain the distribution $q^*(\theta)$ which is the solution of the optimization problem: $\max_q \mathbb{E}_{\theta \sim q}[\eta(\theta)] + \alpha H(q)$, where $H(q) = \mathbb{E}_{\theta \sim q}[-\log q(\theta)]$ is the entropy of $q$. Solving the above equation by setting derivative to zero yields the an energy-based formulation for the optimal policy-parameter distribution: $q^*(\theta) \propto \exp(\frac{\eta(\theta)}{\alpha})$. Drawing samples from this posterior using traditional methods such as MCMC is computationally intractable. Stein variational gradient descent (SVGD; Liu & Wang (2016)) is an efficient method for generating samples and also converges to the posterior of the energy-based model. Let $\{\theta\}_1^n$ be the $n$ particles that constitute the policy ensemble. SVGD provides appropriate direction for perturbing each particle such that induced KL-divergence between the particles and the target distribution $q^*(\theta)$ is reduced. The perturbation (gradient) for particle $\theta_i$ is given by (please see Liu & Wang (2016) for derivation):

$$\Delta\theta_i = \frac{1}{n} \sum_{j=1}^{n} \left[ \nabla_{\theta_j} \log q^*(\theta_j) k(\theta_j, \theta_i) + \nabla_{\theta_j} k(\theta_j, \theta_i) \right]$$

where $k(\theta_j, \theta_i)$ is a positive definite kernel function. Using $q^*(\theta) \propto \exp(\frac{\eta(\theta)}{\alpha})$ as target distribution, and $k(\theta_j, \theta_i) = \exp(-D_{JS}(\rho_{\pi_{\theta_j}}, \rho_{\pi_{\theta_i}})/T)$ as the JS-kernel, we get the gradient direction for ascent:

$$\Delta\theta_i = \frac{1}{n} \sum_{j=1}^{n} \exp(-D_{JS}(\rho_{\pi_{\theta_j}}, \rho_{\pi_{\theta_i}})/T) \left[ \nabla_{\theta_j} \frac{\eta(\pi_{\theta_j})}{\alpha} - \frac{1}{T} \nabla_{\theta_j} D_{JS}(\rho_{\pi_{\theta_j}}, \rho_{\pi_{\theta_i}}) \right]$$

where $\rho_{\pi_\theta}(s, a)$ is the state-action visitation distribution for policy $\pi_\theta$, and $T$ is the temperature. Also, for our case, $\nabla_{\theta_j}\eta(\pi_{\theta_j})$ is the interpolated gradient from self-imitation (Equation 5).

---

[1]https://sites.google.com/site/tesr4t223424

### 5.8.2 Implementation details

The $-\nabla_{\theta_j} D_{JS}(\rho_{\pi_{\theta_j}}, \rho_{\pi_{\theta_i}})$ gradient in the above equation is the repulsion factor that pushes $\pi_{\theta_i}$ away from $\pi_{\theta_j}$. Similar repulsion can be achieved by using the gradient $+\nabla_{\theta_i} D_{JS}(\rho_{\pi_{\theta_j}}, \rho_{\pi_{\theta_i}})$; note that this gradient is w.r.t $\theta_i$ instead of $\theta_j$ and the sign is reversed. Empirically, we find that the latter results in slightly better performance.

**Estimation of** $\nabla_{\theta_i} D_{JS}(\rho_j, \rho_i)$**:** This can be done in two ways - using implicit and explicit distributions. In the implicit method, we could train a parameterized discriminator network ($\phi$) using state-actions pairs from $\pi_i$ and $\pi_j$ to implicitly approximate the ratio $r_{ij}^{\phi} = \rho_{\pi_i}(s,a)/[\rho_{\pi_i}(s,a) + \rho_{\pi_j}(s,a)]$. We could then use the policy gradient theorem to obtain the gradient of $D_{JS}$ as explained in Section 2.2. This, however, requires us to learn $\mathcal{O}(n^2)$ discriminator networks for a population of size $n$, one for each policy pair $(i,j)$. To reduce the computational and memory resource burden to $\mathcal{O}(n)$, we opt for explicit modeling of $\rho_{\pi_i}$. Specifically, we train a network $\rho_{\psi_i}$ to approximate the state-action visitation density for each policy $\pi_i$. The $\rho_{\psi_1} \ldots \rho_{\psi_n}$ networks are learned using the $D_{JS}$ optimization (Equation 2), and we can easily obtain the ratio $r_{ij}(s,a) = \rho_{\psi_i}(s,a)/[\rho_{\psi_i}(s,a) + \rho_{\psi_j}(s,a)]$. The agent then uses $\log r_{ij}(s,a)$ as the SVPG exploration rewards in the policy gradient theorem.

**State-value baselines:** We use state-value function networks as baselines to reduce the variance in sampled policy-gradients. Each agent $\theta_i$ in a population of size $n$ trains $n+1$ state-value networks corresponding to real environment rewards $r(s,a)$, self-imitation rewards $-\log r^{\phi}(s,a)$, and $n-1$ SVPG exploration rewards $\log r_{ij}(s,a)$.

### 5.9 Comparison to Oh et al. (2018)

In this section, we provide evaluation for a recently proposed method for self-imitation learning (SIL; Oh et al. (2018)). The SIL loss function take the form:

$$\mathcal{L}^{SIL} = \mathbb{E}_{s,a,D}\Big[ -\log \pi_\theta(a|s)(R - V_\theta(s))_+ + \frac{\beta}{2}||(R - V_\theta(s))_+||^2\Big]$$

In words, the algorithm buffers $(s,a)$ and the corresponding return $(R)$ for each transition in rolled trajectories, and reuses them for training if the stored return value is higher than the current state-value estimate $V_\theta(s)$.

We use the code provided by the authors [2]. As per our understanding, PPO+SIL does not use a single set of hyper-parameters for all the MuJoCo tasks (Appendix A; Oh et al. (2018)). We follow their methodology and report numbers for the best configuration for each task. This is different from our experiments since we run all tasks on a single fix hyper-parameter set (Appendix 5.5), and therefore a direct comparison of the average scores between the two approaches is tricky.

| | SIL Dense rewards Oh et al. (2018) | SIL Episodic rewards | SIL Noisy rewards Each $r_t$ suppressed w/ 90% prob. ($p_m = 0.9$) | SIL Noisy rewards Each $r_t$ suppressed w/ 50% prob. ($p_m = 0.5$) |
|---|---|---|---|---|
| Walker | 3973 | 257 | 565 | 3911 |
| Humanoid | 3610 | 530 | 1126 | 3460 |
| Humanoid-Standup ($\times 10^4$) | 18.9 | 4.9 | 14.9 | 18.8 |
| Hopper | 1983 | 563 | 1387 | 1723 |
| Swimmer | 120 | 17 | 50 | 100 |
| InvertedDoublePendulum | 6250 | 405 | 6563 | 6530 |

Table 3: Performance of PPO+SIL (Oh et al., 2018) on tasks with episodic rewards, noisy rewards with masking probability $p_m$, and dense rewards. All runs use 5M timesteps of interaction with the environment.

Table 3 shows the performance of PPO+SIL on MuJoCo tasks under the various reward distributions explained in Section 3.1 - dense, episodic and noisy. We observe that, compared to the dense rewards setting (default Gym rewards), the performance suffers under the episodic case and when the rewards are masked out with $p_m = 0.9$. Our intuition is as follows. PPO+SIL makes use of the cumulative

---

[2]https://github.com/junhyukoh/self-imitation-learning

return ($R$) from each transition of a past good rollout for the update. When rewards are provided only at the end of the episode, for instance, cumulative return does not help with the temporal credit assignment problem and hence is not a strong learning signal. Our approach, on the other hand, derives dense, per-timestep rewards using an objective based on divergence-minimization. This is useful for credit assignment, and as indicated in Table 1. (Section 3.1) leads to learning good policies even under the episodic and noisy $p_m = 0.9$ settings.

## 5.10 Comparison to off-policy RL (Q-learning)

Our approach makes use of replay memory $\mathcal{M}_E$ to store the past good rollouts of the agent. Off-policy RL methods such as DQN (Mnih et al., 2015) also accumulate agent experience in a replay buffer and reuse them for learning (e.g. by reducing TD-error). In this section, we evaluate the performance of one such recent algorithm - Twin Delayed Deep Deterministic policy gradient (TD3; Fujimoto et al. (2018)) on tasks with episodic and noisy rewards. TD3 builds on DDPG (Lillicrap et al., 2015) and surpasses its performance on all the MuJoCo tasks evaluated by the authors.

| | TD3 Dense rewards Fujimoto et al. (2018) | TD3 Episodic rewards | TD3 Noisy rewards Each $r_t$ suppressed w/ 90% prob. ($p_m = 0.9$) | TD3 Noisy rewards Each $r_t$ suppressed w/ 50% prob. ($p_m = 0.5$) |
|---|---|---|---|---|
| Walker | 4352 | 189 | 395 | 2417 |
| Hopper | 3636 | 402 | 385 | 1825 |
| InvertedDoublePendulum | 9350 | 363 | 948 | 4711 |
| Swimmer* | - | - | - | - |
| Humanoid-Standup* | - | - | - | - |
| Humanoid* | - | - | - | - |

Table 4: Performance of TD3 (Fujimoto et al., 2018) on tasks with episodic rewards, noisy rewards with masking probability $p_m$, and dense rewards. All runs use 5M timesteps of interaction with the environment.

Table 4 shows that the performance of TD3 suffers appreciably with the episodic and noisy $p_m = 0.9$ reward settings, indicating that popular off-policy algorithms (DDPG, TD3) do not exploit the past experience in a manner that accelerates learning when rewards are scarce during an episode.

\* For 3 tasks used in our paper—Swimmer and the high-dimensional Humanoid, Humanoid-Standup—the TD3 code from the authors [3] is unable to learn a good policy even in presence of dense rewards (default Gym rewards). These tasks are also not included in the evaluation by Fujimoto et al. (2018).

## 5.11 Comparing SVPG exploration to a novelty-based baseline

We run a new exploration baseline - EX$^2$ (Fu et al., 2017) and compare its performance to SI-interact-JS on the hard exploration MuJoCo tasks considered in Section 3.2. The EX$^2$ algorithm does implicit state-density $\rho(s)$ estimation using discriminative modeling, and uses it for novelty-based exploration by adding $-\log \rho(s)$ as the bonus. We used the author provided code [4] and hyperparameter settings. TRPO is used as the policy gradient algorithm.

| | EX$^2$ | SI-interact-JS |
|---|---|---|
| SparseHalfCheetah | -286 | 769 |
| SparseHopper | 1477 | 1949 |
| SparseAnt | -3.9 | 208 |

Table 5: Performance of EX$^2$ (Fu et al., 2017) and SI-interact-JS on the hard exploration MuJoCo tasks from Section 3.2. SparseHalfCheetah, SparseHalfCheetah, SparseAnt use 1M, 1M and 2M timesteps of interaction with the environment, respectively. Results are averaged over 3 separate runs.

---

[3] https://github.com/sfujim/TD3
[4] https://github.com/jcoreyes/ex2

