# OpenReview forum: "Learning Self-Imitating Diverse Policies"
_ICLR.cc/2019/Conference_

### Official Review · AnonReviewer1 · 2018-10-31
**Good paper, accept**

**Rating:** 8
**Confidence:** 4

**Review:**

Overall impression:
I think that this is a well written interesting paper with strong results. One thing I’d have liked to see a bit more is an explanation of why self imitation is more effective than standard policy gradient? Where does the extra supervision/stability come from, and can this be explained intuitively? I’ve suggested some small changes/clarifications to be made inline, and a few more comparisons to add. But overall, I very much like this line of work and I recommend accepting this paper.


Abstract:
We demonstrate its effectiveness on a number of challenging tasks. -> be more specific.

The term single-timestep optimization is not very clear. Can this be clarified?

they are more widely applicable in the sparse or episodic reward settings -> it is likely important to mention that they are agnostic to horizon of the task.

Related works:
Guided Policy Search also does divergence minimization. GAIL considers the imitation learning work as a sort of divergence minimization problem as well, which should be explicitly mentioned. Other work for good exploration include DIAYN (Eysenbach et al 2018). The difference in resulting updates between (Oh et al) and this work should be clearly discussed in the methods section.

“we learn shaped, dense rewards”-> too early in the paper for this to make sense. can provide some contextt

Section 2.2:
fully decides the expected return -> clarify this a bit. I think what you mean is that the dynamics are wrapped into this already, so it accounts for this, but this can be made explicit.

Small typos in appendix 5.1 (r should be replaced by the density ratio)

The update in (3) seems quite similar to what GAIL would do. What is the difference there? Or is the difference just in the fact that the experts are chosen from “self” experiences.

How is the priority list threshold and size chosen?
 Would a softer version of the priority queue update do anything useful? Or would it just reduce to policy gradient when weighted by rewards?

Appendices are very clear and very informative while being succinct!

I would have liked to see Appendix 5.3 in the main text (maybe a shorter form) to clarify the whole algorithm

What is psi in appendix 5.3? The algorithm remains a bit unclear without this clarification

Experiments.
Only 1 question to answer in this section is labelled? Put 2) and 3) appropriately.

Can a comparison to Oh et al 2018 be added to this for the sake of completeness? Also can this be compared to using novelty/curiosity based exploration schemes?

Can the authors comment on why the method reaches higher asymptotic performance but is often slower in the beginning than the other methods in Fig 3.

---

> ### Author Response · Authors · 2018-11-26
> **Response to AnonReviewer1. Thank you for your comments! (Part 1/2)**
>
>
> 1- Concerning “Why is self-imitation more effective than standard policy gradients, and if the source of stability can be explained intuitively” :
>
> We believe that learning pseudo-rewards with self-imitation helps in the temporal credit assignment problem in the sparse- or episodic-reward setting. For instance, in the episodic setting, where a reward is only provided at episode termination, standard policy gradient algorithms reinforce the actions towards the beginning of the episode based on a reward signal which is obtained after multiple timesteps and convolves the effect of many intermediate actions. This signal is potentially sparse and diluted, and may deteriorate with task horizon. With our approach, since we learn “per-timestep” pseudo-rewards with self-imitation, we expect this greedy signal to help in attributing credit to actions more effectively, leading to faster training.
>
> Qualitatively, the stability of the self-imitation algorithm could also be understood by viewing it as a form of curriculum learning [4]. Unlike learning from perfect demonstrations by external experts, our learner at any point in time is imitating only a slightly different version of itself. The demonstrations, therefore, increase in complexity gradually over time, resulting in an implicit, adaptive curriculum which stabilizes learning and avoids catastrophic forgetting of behaviors.
>
> [4] Bengio, Yoshua, et al. "Curriculum learning." Proceedings of the 26th annual international conference on machine learning. ACM, 2009.
>
>
> 2- Concerning “Re-phrases in various sections”:
>
> We have incorporated all the suggested changes in the revision with extra discussion. We have also added the missing reference to Guided Policy Search and expanded on GAIL. DIAYN (Eysenbach et al 2018) is included in Appendix 5.6.
>
>
> 3- Concerning “Comparison to Oh et al. (2018)”:
>
> We have added a new section (Appendix 5.9) focussed on the algorithm (SIL) by Oh et al. (2018). Therein, we mention the update rule for SIL and the performance of PPO+SIL on MuJoCo tasks under the various reward distributions we used in our paper. We summarize our observations here (please see Appendix 5.9 for more details). The performance of PPO+SIL suffers under the episodic case and when the rewards are masked out with 90% probability (p_m=0.9). Our intuition is that this is because PPO+SIL makes use of the “cumulative return” from each transition of a past good rollout for the update. When rewards are provided only at the end of the episode, for instance, cumulative return does not help with the temporal credit assignment problem and hence is not a strong learning signal.
>
>
> 4- Concerning “Comparing SVPG exploration (Figure 3) to novelty/curiosity based exploration schemes”:
>
> We have added a new section (Appendix 5.11) on comparing SVPG exploration to a novelty-based exploration baseline - EX2 [3]. The EX2 algorithm does implicit density estimation using discriminative modeling, and uses it for novelty-based exploration. We report results on the hard exploration MuJoCo tasks considered in Section 3.2, using author provided code and hyperparameters. Table 5 in Appendix 5.11 shows that we compare favorably against EX2 on the tasks evaluated.
>
> [3] Fu, Justin, John Co-Reyes, and Sergey Levine. "Ex2: Exploration with exemplar models for deep reinforcement learning." Advances in Neural Information Processing Systems. 2017.
>
>
> 5- Concerning “What is psi in appendix 5.3? ”:
>
> We apologize for skimping the details on this. “psi” denotes the parameters of neural networks that are used to model the state-action visitation distribution (rho) of the policy. Therefore, for an ensemble of n policies, there are n “psi” networks. The motivation behind using these networks is as follows. To calculate the gradient of JS, we need the ratio denoted by r^{\phi} in the paper. This ratio can be obtained implicitly by training a parameterized discriminator network. However, when using SVPG exploration with JS kernel, this method would require us to train O(n^2) discriminator networks, one each for calculating the gradient of JS between a policy pair (i,j). To reduce the computational and memory resource burden to O(n), we opt for explicit modeling of the state-action visitation distribution (rho) of the policy by a network with parameters “psi”. The “psi” networks are trained using the JS optimization (Equation 2.) and we can then obtain the ratio explicitly from these “psi” networks. We have added these details (and more) to Appendix 5.8.2. It also contains proper symbols (in Latex) for easier reading.

---

> > ### Author Response · Authors · 2018-11-26
> > **Response to AnonReviewer1. Thank you for your comments! (Part 2/2)**
> >
> >
> > 6- Concerning “How is the priority list threshold and size chosen?”:
> >
> > Our implementation stores the top-C trajectories in the priority queue based on cumulative trajectory-return. We fix the capacity (C) to 10 trajectories for all our experiments. This number was chosen after a limited hyperparameter grid search on Humanoid and Hopper (Appendix 5.4). In general, we didn’t find our method to be particularly sensitive to the choice of C.
> >
> >
> > 7- Concerning “Would a softer version of the priority queue update do anything useful?”:
> >
> > In our initial experiments, we tested with using more relaxed update rules for the priory queue, but found that storing the overall top-C trajectories gave the best results. Nonetheless, the various options for storing and reusing past experiences present interesting trade-offs, and we hope to look deeper into this in the future.
> >
> >
> > 8- Concerning “The update in (3) seems quite similar to what GAIL would do. What is the difference there?”:
> >
> > Yes, as we mention in the derivation (Appendix 5.1), GAIL does a similar update, but using external expert trajectories rather than using self-imitation. An implementation-specific difference is that while GAIL uses discriminator networks to implicitly estimate the ratio required in the policy gradient theorem, we (when using SVPG exploration in Algorithm 2) learn separate state-action density estimation networks (psi), and explicitly compute the required ratios. This is done for reasons of computational efficiency (Appendix 5.8.2).
> >
> >
> > 9- Concerning “why higher asymptotic performance but is often slower in the beginning than the other methods in Fig 3”:
> >
> > Consider SparseHopper as an example. There is a local minima where the agent can stand still (i.e. no hopping) and collect the per-timestep survival bonus given for not falling down. Baseline algorithms such as PPO-Independent or SI-independent quickly get into this local minima since they greedily exploit the survival bonus readily available. Hence, they reach a score of ~1000 quickly. In, SI-Interact-JS, however, the JS repulsion forces the agents to be diverse and explore the state-space much more effectively. The highest scoring agent in this ensemble (which is plotted in Figure 3.) discovers the hopping behavior eventually. However, during its learning lifetime, it takes varied actions to reach states different from other agents, due to JS repulsion. The score grows gradually since many of the attempts in the beginning lead to the agent falling down (and therefore episode termination) in the process of trying something different. The agent does not quickly accumulate the survival bonus and stand still, unlike the baselines. The asymptotic score is higher since the forward hopping is rewarded higher compared to the survival bonus.

---

### Official Review · AnonReviewer3 · 2018-11-04
**Well written paper that explores an interesting idea, weak experimental evaluation**

**Rating:** 6
**Confidence:** 2

**Review:**

The paper proposes how previously experienced high reward trajectories can be used to generate dense reward functions for more for efficient training of policies in context of reinforcement learning. The paper does this by computing the state-action pair distribution of high rewarding trajectories in the replay buffer, and using a surrogate reward that measures the distance between this distribution and the current state-action pair distribution. The paper derives approximate policy gradients for this surrogate reward function. The paper then describes limitations of doing this: possibility of getting stuck in the local neighborhood of currently well-performing trajectories. It also describes an extension based on Stein variational policy gradients to diversify behavior of an ensemble of policies that are learned together. The paper shows experimental results on a number of MuJoCo tasks.

Strengths:
1. Adequately leveraging high-return roll-outs for effective learning of policies is an important problem in RL. The paper proposes and empirically investigates a reasonable approach for doing this. The paper shows how using the proposed additional rewards leads to better performance on the choses benchmarks than baseline methods without the proposed rewards.

2. I also like that the paper details the short-comings of the proposed approach, and how these could be fixed.

Weaknesses:
1. The paper uses sparse rewards in RL as a motivation. However, the proposed approach crucially relies on the fact that a good trajectory has at least been encountered once in the past to be of any use. I am not sure if how the proposed approach does justice to the motivation in the paper. The paper should re-write the motivation, or better explain why the proposed method addresses the motivation.

2. Additionally, the paper does not provide adequate experimental validation. The experiment that I think will make the case for the paper is one that shows the sample efficiency of the proposed approach over other baseline methods, when given a successful past roll-out. The current experimental setup emphasizes the sparse reward scenario in RL, and it is just not clear to me as to why this is a good benchmark to study the effects of the proposed method.

3. The paper primarily makes comparisons to on-policy methods. This may not be a fair comparison, as the proposed method uses past trajectories from a replay buffer (to compute reward). Perhaps improvements are coming because of use of this off-policy information. The paper should design experiments to de-conflate this: perhaps by also comparing to how these additional rewards will compare in context of off-policy methods (like Q-learning).

4. I also do not understand how the benchmark tasks were chosen? Are the MuJoCo tasks studied here a fair representative of MuJoCo tasks studied in literature, or are these selected in any manner? While selecting and modifying benchmarks for the purpose of making a specific point is acceptable, it is important to include benchmark results on a full suite of tasks. This can help understand (desirable or un-desirable) side-effects of proposed ideas.

After reading author response and the extra experiments, I have changed my rating to 6 (from the original rating of 5).

---

> ### Author Response · Authors · 2018-11-26
> **Response to AnonReviewer3. Thank you for your comments!**
>
>
> 1- Concerning “Points 1. and 2. under Weaknesses” :
>
> We do not wish to claim or motivate that self-imitation would suffice if the task is “sparse” in the sense that most of the episodes don’t see *any* rewards. This would fall under the limitations of self-imitation which we discuss in the paper; we could rely on population-based exploration methods (e.g. SVPG, Section 2.3) and draw on the rich literature on single-agent exploration methods like curiosity/novelty-search or parameter noise to alleviate this to an extent. Instead, we focus on scenarios where “sparse” feedback is available within an episode. We will make this very clear in our revision. For example, our experiments in Section 3.1 consider tasks where some feedback is available in an episode - either only once at the end of the episode, or at very few timesteps during an episode. We find self-imitation to be highly beneficial (compared to standard policy gradients) on these “sparse” constructions. Some practical situations of the kind include a.) robotics tasks where rewards in an episode could be intermittent or delayed by arbitrary timesteps due to the inverse kinematics operations b.) cases where a mild feedback on the overall quality of the episode is available, but designing a dense reward function manually is prohibitively hard; an interesting example of this is [5].
>
> Also, although our algorithm exploits “good” trajectories from agent’s past experience, the demands on the “goodness” of the trajectories are very relaxed. Indeed, the trajectories imitated during the initial phases of learning have quite low overall scores, and they gradually improve in quality.
>
> [5] Christiano, Paul F., et al. "Deep reinforcement learning from human preferences." Advances in Neural Information Processing Systems. 2017.
>
>
> 2- Concerning “Point 3. under Weaknesses -- comparison to off-policy RL methods”:
>
> Our approach makes use of a replay memory to store and exploit past good rollouts of the agent. Off-policy RL methods such as DQN, DDPG also accumulate agent experience in a replay buffer and reuse them for learning (e.g. by reducing TD-error). We run new experiments with a recent off-policy RL method based on DDPG - Twin Delayed Deep Deterministic policy gradient (TD3; [2]). Appendix 5.10 evaluates its performance on MuJoCo tasks under the various reward distributions we used in our paper. We find that the performance of TD3 suffers appreciably under the episodic case and when the rewards are masked out with 90% probability (p_m=0.9). We therefore believe that popular off-policy algorithms (DDPG, TD3) do not exploit the past experience in a manner that accelerates learning when rewards are scarce during an episode. The per-timestep (dense) pseudo-rewards that we obtain with the divergence-minimization objective help in temporal credit assignment, resulting in good policies even under the episodic and noisy (p_m=0.9) settings (Table 1, Section 3.1).
>
> [2] Fujimoto, Scott, Herke van Hoof, and Dave Meger. "Addressing Function Approximation Error in Actor-Critic Methods." International Conference on Machine Learning. 2018.
>
>
> 4- Concerning “Point 4. under Weaknesses ”:
>
> We have added Appendix 5.7 with results on more MuJoCo tasks. Combined with Table 1. in the paper, we believe our overall set to be fairly representative. For reference, the PPO paper [6], which forms our baseline, uses the same set of benchmarks (Figure 3 in their paper).
>
> [6] Schulman, John, et al. "Proximal policy optimization algorithms." arXiv preprint arXiv:1707.06347 (2017).

---

> > ### Comment · AnonReviewer3 · 2018-12-05
> > **Thank You!**
> >
> > Thank you for providing a detailed reply. I hope authors will incorporate these points into the paper (specifically the results on a more comprehensive benchmark suite (my concern in my 4th point).
> >
> > I also hope authors will release code and scripts to reproduce the results in the paper, so as to make future comparisons possible.

---

> > > ### Author Response · Authors · 2018-12-06
> > > **Thank you for the updated rating.**
> > >
> > > We would merge pieces from the Appendix into the main sections for better coherence. Also, we would make our source code and scripts public.

---

> > > > ### Public Comment · ~Yuchen_Lu1 · 2019-03-08
> > > > **On the status of source code**
> > > >
> > > > It's a very interesting approach. Is there still any plan on releasing the source code?

---

> > > > > ### Author Response · Authors · 2019-03-11
> > > > > **Code to be released soon.**
> > > > >
> > > > > Hi,
> > > > > Thanks for your interest in our paper! We are in the process of cleaning up the code for release. We expect it to be ready by the end of this month.

---

### Official Review · AnonReviewer2 · 2018-11-05
**intuitive/elegant idea, well-written, convincing results**

**Rating:** 8
**Confidence:** 3

**Review:**

The paper describes a method to improve reinforcement learning for task with sparse rewards signals.

The basic idea is to select the best episodes from the system's experience, and learn to imitate them step by step as the system evolves, aiming at providing a less sparse learning signal.

The math works out to a gradient that is of similar form as a policy gradient, which makes it easy to interpolate both of them. The resulting training procedure is a policy gradient that gets additional reinforcement of the system's best runs.

The experiments show the validity especially for the most extreme case (episodic rewards), while, as expected, for the other extreme of dense rewards, the method's effect is not consistently positive.

The paper then critiques its own method and identifies a critical weakness: the reliance on good exploration. I like that a lot. The paper goes on to suggest an extension to address this by training an ensemble, and shows the effectiveness of this for a number of tasks. However, I feel that the description of this extension is less clear than that of the core idea, and introduces too many new ideas and concepts in a too condensed text.

The paper seems a significant in that it provides a notable improvement for sparse-rewards tasks, which are a common sub-class of real-world problems.

My background is not RL. While I am quite confident in my understanding of the paper's math, I am not 100% familiar with the typical benchmark sets. Hence, I cannot judge whether the results include good baselines, or whether the task selection is biased. I can also not judge the completeness of the related work, and how novel the work is. For these questions, I hope that the other reviewers can provide more information.

Pros:
 - intuitive idea for a common problem
 - solution elegantly has the form of a modified policy gradient
 - convincing experimental results
 - self-critique of core idea, and extension to address its main weakness
 - nicely written text, does not leave a lot of questions

Cons:
 - while the core idea is nicely motivated and described and good to follow, Section 2.3 feels very dense and too short.

Overall, I find the core idea quite intuitive and elegant. The paper's background, motivation, and core method are well-written and, with some effort, quite readable for someone who is not an RL expert. I found that several questions I had during reading were preempted promptly and addressed. However, the description of the secondary method (Section 2.3) is too dense.

To me, the paper solidly meets the threshold of publication. Since I have no good comparison to other papers, I rate it a "clear accept" (8).

Minor points:

I noticed a few superfluous "the", please double-check.

In Table 1, please use the same exponent for directly comparable numbers, e.g. instead of "1.8e5 4.4e4", say "18e4 4.4e4". Or best just print the full numbers without exponent, I think you have the space.

When reading Table 1, I could bnot immediately line up "PPO" and "Self-imitation" in the caption with the table columns. It took a while to infer that PPO refers to \nu=0, and SI to \nu=0.8. Can you add PPO and SI to the table headings?

You define p as "the masking probability", but it is not clear whether that is the probability for keeping a "1" in the mask,
or for masking out the value. I can only guess from the results. I suggest to rephrase as "the probability of retaining a reward". Also, how about using plain words in Table 1's heading, such as "Noisy rewards\nSuppressing 10% of rewards", so that one can understand the table without having to search for its description in the text?

---

> ### Author Response · Authors · 2018-11-26
> **Response to AnonReviewer2. Thank you for your comments!**
>
>
> 1- Concerning “Section 2.3 being too dense” :
>
> We have re-organized the writing. Specifically, we have added more details on SVPG exploration with the JS-kernel in Appendix 5.8. Appendix 5.8.1 includes some more intuition and theory behind Stein Variational Gradient Descent (SVGD) and Stein Variational Policy Gradient (SVPG). Appendix 5.8.2 contains details on our implementation such as calculation of SVPG exploration rewards by each agent, and state-value function baselines, along with better explanation of symbols used in our full algorithm (Algorithm 2).
>
> 2- Concerning “Minor points”:
>
> Thank you for pointing these out. We have changed Table 1. in the revision to include all the suggested changes, in the hope that the table becomes self-explanatory. We have also rephrased the text to clarify that we compare performance with two different reward masking values - suppressing each per-timestep reward r_t with 90% probability (p_m = 0.9), and with 50% probability (p_m=0.5).

---

### Public Comment · (anonymous) · 2018-11-16
**How to Calculate JS kernel**

I enjoyed reading your interesting submission, and I have one question about implementation.

How did you calculate JS kernel, k(theta_j , theta_i)=exp(-D_JS(rho_pi_theta_i, rho_pi_theta_j)/T)?

I think in order to calculate D_JS(rho_pi_theta_i, rho_pi_theta_j), we have to train discriminators which differentiate between trajectories from rho_pi_theta_i and  trajectories from rho_pi_theta_j. If this thought is right, we have to 28 discriminators for all combinations of 8 policies. However, this is not practical.

If replay memory is shared, D_JS can be calculated by using 2 discriminators, r^phi_i and r^phi_j. This is because rho_pi_theta_i/rho_pi_theta_j = rho_pi_theta_i/rho_pi_E * rho_pi_theta_E/rho_pi_theta_j = r^phi_i / (1-r^phi_i)  * (1 -r^phi_j)/r^phi_j . However, in your paper, replay memories are not shared.

Therefore, I would like to know how to calculate JS kernel.

Thank you!!

---

> ### Author Response · Authors · 2018-11-26
> **Thank you for your question and interest in our paper.**
>
>
> You are correct in observing that if we use parameterized discriminator networks to estimate the ratio $r^{\phi}_{ij} = \rho_{\pi_i}(s,a) / [\rho_{\pi_i}(s,a) + \rho_{\pi_j}(s,a)]$ for the SVPG exploration rewards, then we would need O(n^2) discriminator networks, for n policies in the ensemble. To ensure scalability to ensembles of large number of policies, we opt for explicit modeling of the state-action visitation density for each policy (i) by a parameterized network $\psi_i$. With this, we can obtain the ratios for the SVPG exploration rewards using the n $\psi$ network, reducing the complexity to O(n). Please check the recently added Appendix 5.8.2 in our revision for more details. We would be happy to answer any further questions you may have on this.

---

> > ### Public Comment · (anonymous) · 2018-12-06
> > **Whether discriminator or $\psi$ network did you use? and about hyparas**
> >
> > Thank you for your reply!
> > And I have two more questions.
> >
> > 1. Whether discriminator or $\psi$ network did you use for getting results you write in  your paper?
> > 2. What number you use for $\alpha$ for SVPG and T for JS kernel?
> >
> > Thank you!

---

> > > ### Author Response · Authors · 2018-12-06
> > > **Implementation Details**
> > >
> > >
> > > 1. Experiments in section 3.1 use a parameterized discriminator since a single network suffices for self-imitation. Experiments in section 3.2 use $\psi$ networks for computational efficiency with policy ensembles.
> > >
> > > 2. In practice, to get the complete SVPG gradient, we calculate the exploitation and exploration components, and then do a convex combination as: (1-p)*exploitation + p*exploration, where p is linearly decayed from 1 to 0. The temperature (T) is held constant at 0.5 (2D-navigation) and 0.2 (Locomotion).

---

### Author Response · Authors · 2018-11-26
**General response to the reviewers**

We would like to thank the anonymous reviewers for their comments and constructive feedback. We address each reviewer's comments individually and summarize the major additions to the revision here:

1. Added Appendix 5.7 with results on more MuJoCo tasks
2. Added Appendix 5.8 with SVPG background and our implementation details.
3. Added Appendix 5.9 on comparison to Oh et al. (2018) [1]
4. Added Appendix 5.10 on comparison to off-policy RL (TD3, Fujimoto et al. (2018)) [2]
5. Added Appendix 5.11 on comparing SVPG exploration to a novelty-based baseline (EX^2, Fu et al. (2017)) [3]

[1] Oh, Junhyuk, Yijie Guo, Satinder Singh, and Honglak Lee. "Self-Imitation Learning." International Conference on Machine Learning. 2018.
[2] Fujimoto, Scott, Herke van Hoof, and Dave Meger. "Addressing Function Approximation Error in Actor-Critic Methods." International Conference on Machine Learning. 2018.
[3] Fu, Justin, John Co-Reyes, and Sergey Levine. "Ex2: Exploration with exemplar models for deep reinforcement learning." Advances in Neural Information Processing Systems. 2017.

---

### Meta-Review · Area_Chair1 · 2018-12-13

**Confidence:** 4
**Recommendation:** Accept (Poster)

**Metareview:**

This paper proposes a reinforcement learning approach that better handles sparse reward environments, by using previously-experienced roll-outs that achieve high reward. The approach is intuitive, and the results in the paper are convincing. The authors addressed nearly all of the reviewer's concerns. The reviewers all agree that the paper should be accepted.